

# Local-scale determinants of arboreal spider beta diversity in a temperate forest: roles of tree architecture, spatial distance, and dispersal capacity

Qiongdao Zhang[1], Dong He[2], Hua Wu[3], Wei Shi[1] and Cong Chen[4]

[1] State Key Laboratory of Biocontrol, Sun Yat-sen University, Guangzhou, China
[2] School of Ecological and Environmental Science, East China Normal University, Shanghai, China
[3] Guangdong Key Laboratory of Integrated Pest Management in Agriculture, Guangdong Institute of Applied Biological Resources, Guangzhou, China
[4] Co-Innovation Center for the Sustainable Forestry in Southern China, College of Forestry, Nanjing Forestry University, Nanjing, China

## ABSTRACT

Spiders are a functionally important taxon in forest ecosystems, but the determinants of arboreal spider beta diversity are poorly understood at the local scale. We examined spider assemblages in 324 European beech (*Fagus sylvatica*) trees of varying sizes across three forest stands in Würzburg (Germany) to disentangle the roles of tree architecture, spatial distance, and dispersal capacity on spider turnover across individual trees. A large proportion of tree pairs (66%) showed higher compositional dissimilarity in spider assemblages than expected by chance, suggesting prominent roles of habitat specialization and/or dispersal limitation. Trees with higher dissimilarity in DBH and canopy volume, and to a lesser extent in foliage cover, supported more dissimilar spider assemblages, suggesting that tree architecture comprised a relevant environmental gradient of sorting spider species. Variation partitioning revealed that 28.4% of the variation in beta diversity was jointly explained by tree architecture, spatial distance (measured by principal coordinates of neighbor matrices) and dispersal capacity (quantified by ballooning propensity). Among these, dispersal capacity accounted for a comparable proportion as spatial distance did (6.8% vs. 5.9%). Beta diversity did not significantly differ between high- and low-vagility groups, but beta diversity in species with high vagility was more strongly determined by spatially structured environmental variation. Altogether, both niche specialization, along the environmental gradient defined by tree architecture, and dispersal limitation are responsible for structuring arboreal spider assemblages. High dispersal capacity of spiders appears to reinforce the role of niche-related processes.

# INTRODUCTION

Beta diversity, defined as the compositional dissimilarity between species assemblages, is an important component of biodiversity (*Whittaker, 1960*; *Anderson et al., 2011*). Beta diversity considers valuable information about how biodiversity is spatially organized and

Corresponding author
Qiongdao Zhang,
zhangqd3@mail2.sysu.edu.cn,
allenzqd520@163.com

provides a link between local and regional patterns, thereby serving as a key conceptual tool for understanding species distributions and community assembly (*Anderson et al., 2011*). Furthermore, exploring the patterns and causes of beta diversity could provide valuable insights for biodiversity conservation and management (*Socolar et al., 2016*). To date, beta diversity studies have been largely biased towards plants and birds (*Socolar et al., 2016*), whereas spiders, another functionally important taxon, have been relatively less studied in this respect (*Carvalho et al., 2011*; *Rodriguez-Artigas, Ballester & Corronca, 2016*).

Spiders are a functionally important taxon in terrestrial ecosystems (*Riechert, 1974*; *Wise, 1993*). They consume a large variety of prey and respond readily to vegetation changes (*Riechert, 1974*; *Riechert & Lockley, 1984*; *Gómez, Lohmiller & Joern, 2016*). Therefore, spiders may serve as a useful indicator of environmental quality and overall biodiversity in an ecosystem (*Willett, 2001*). In forests, spiders can achieve high levels of local species richness and abundance from the litter layers to the canopy (*Basset, 2001*). As such, it is important to identify the determinants of arboreal spider diversity to better understand the mechanism of spider community assembly and inform biological conservation in forests. Previous studies have suggested that environmental heterogeneity, spatial distance and/or dispersal capacity differences can be relevant for beta diversity in general (*Jiménez-Valverde et al., 2010*; *Carvalho et al., 2011*; *Rodriguez-Artigas, Ballester & Corronca, 2016*). However, the influence of these factors on beta diversity of arboreal spiders is still unclear.

According to niche theory, which assumes habitat specialization as an important driver of species turnover, environmental heterogeneity is the most relevant predictor of beta diversity (*Gilbert & Lechowicz, 2004*; *Legendre et al., 2009a*). For spiders, structural complexity of vegetation, which affects the availability of web attachment structures (*Rypstra et al., 1999*) and favored prey (*Harwood, Sunderland & Symondson, 2003*), has been recognized as an important environmental factor influencing their community composition and diversity at the stand level (*Halaj, Ross & Moldenke, 1998*; *Halaj, Ross & Moldenke, 2000*; *Jiménez-Valverde & Lobo, 2007*; *Carvalho et al., 2011*; *Rodriguez-Artigas, Ballester & Corronca, 2016*). However, the influence of individual plant architecture on arboreal spider beta diversity is poorly understood. An investigation of this influence may help elucidate lower-level processes regulating diversity. Since trees, especially their canopies, provide arboreal spiders with sites for web attachment, foraging, oviposition, and shelter (*Halaj, Ross & Moldenke, 2000*; *Hsieh, 2011*), the architecture of individual tree may serve as the immediate habitat template for spider assemblages in tree canopies (*Halaj, Ross & Moldenke, 2000*). Hence, we predict that tree pairs with a more similar architecture will support a more similar species composition.

By definition, beta diversity integrates the spatial dimension, thus spatial distance is also proposed as an important predictor of beta diversity (*Soininen, McDonald & Hillebrand, 2007*). With the rise of neutral theory, which highlights the role of dispersal limitation (*Hubbell, 2001*), the importance of spatial distance in predicting species distribution is increasingly recognized. Because spatial and environmental factors usually interact with each other, much effort (*Tuomisto, Ruokolainen & Yli-Halla, 2003*; *Anderson et al., 2011*) has been invested towards partitioning the variation in species beta diversity into pure spatial component, pure environmental component and spatially structured environmental
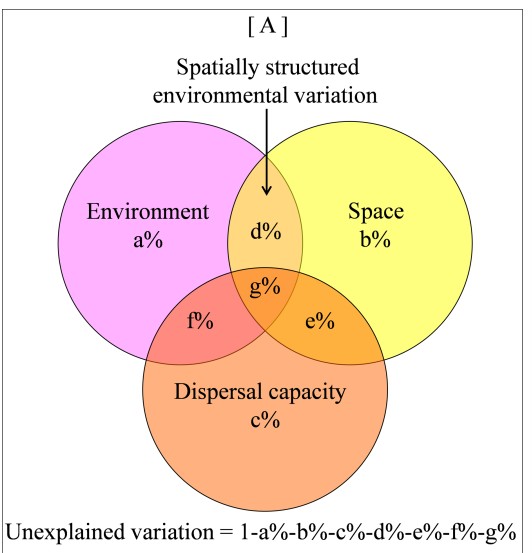
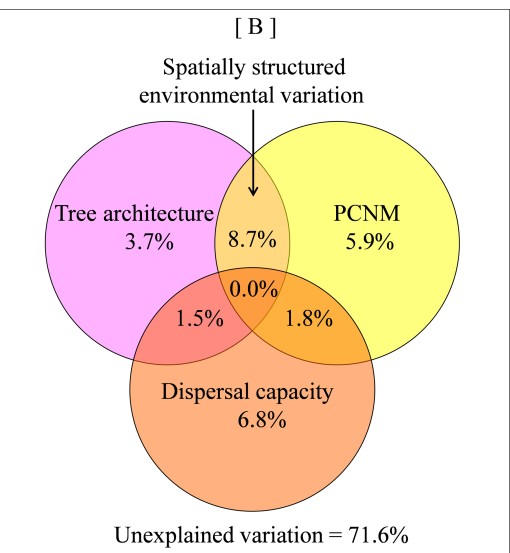

**Figure 1  Venn diagram showing relative influence of tree architecture, spatial distance and dispersal capacity on beta diversity.** (A) a conceptual representation of the contrast among environment, space, dispersal capacity and spatially-structured environmental components; (B) empirical variance partitioning of tree architecture, spatial distance, dispersal capacity effects on spider beta diversity.

component (i.e., the interaction between space and environments, Fig. 1A). The distinction between spatial and environmental influences on diversity is of paramount importance to community assembly studies because it provides insight into the underlying processes that create patterns of beta diversity (*Legendre et al., 2009a*; *Jiménez-Valverde et al., 2010*; *Carvalho et al., 2011*).

The relative contribution of spatial distance to beta diversity often indicates the importance of dispersal limitation (*Jiménez-Valverde et al., 2010*), but dispersal capacity is actually not equal across species. Dispersal limitation is more likely a result from the interaction between spatial distance and dispersal capacity. Therefore, it would be helpful to disentangle the relative roles of spatial distance and dispersal capacity in shaping beta diversity. While dispersal capacity is difficult to quantify in many taxa (*Thomas, Brain & Jepson, 2003*; *Werth et al., 2006*; *Engler et al., 2009*), it has been shown to well relate to ballooning propensity in spiders (*Thomas, Brain & Jepson, 2003*; *Jiménez-Valverde et al., 2010*). Variation in body size and ecological strategy among spider species results in a high diversity of ballooning propensity, a characteristic passive air dispersal using silk threads (*Dean & Sterling, 1985*; *Richardson et al., 2006*). We hence expect that ballooning propensity has a comparative contribution to spider beta diversity as spatial distance.

Furthermore, the distribution of a highly mobile species is likely more dependent on environmental than on spatial factors, as these spiders are able to move from unfavorable to favorable habitats. A less mobile species may be more restrained by spatial distance because of their inability to disperse to distant optimal environments (*Araújo & Pearson, 2005*). If a more mobile species coincides with a wide-occupancy species, we predict that

 

spatially structured environmental variables will be a major determinant of spider beta diversity for the more mobile species.

The present study focuses on spider beta diversity pattern in a temperate deciduous forest in Germany, which is dominated by European beech (*Fagus sylvatica*). Specifically, we attempted to answer the following questions: (i) How does individual tree architecture modify arboreal spider beta diversity? (ii) What is the relative importance of tree architecture, spatial distance and dispersal capacity on beta diversity? (iii) Do spiders with varying dispersal capacity show different beta diversity patterns?

## MATERIALS AND METHODS

### Study area

We used a dataset collected by *Hsieh (2011)* at the Würzburg University Forest (50°01′N, 10°30′E) in Germany. The forest covers about 2,664 ha, 18% of which is classified as "High Conservation Value Forest" by the Forest Stewardship Council. Mean yearly temperature and precipitation are 7.5 °C and 675 mm, respectively, in this region. European beech (*F. sylvatica,* nearly 30% of abundance) is the most common tree species inthe study area, followed by oak (*Quercusrobur*, 19%) and spruce (*Piceaabies*, 12%). We focused on beech trees to investigate the composition, diversity and species turnover of arboreal spiders.

Three forest stands with varying age and height characteristics were surveyed: old-growth beech (>150 years old, 20–26 m tall), mature beech (50–60 years old, 13–19 m tall), and young beech (20–25 years old, 5–6 m tall). These three forest stands were distributed in proximity within a 6-ha area. Within each stand, 108 trees were selected to sample arboreal spider assemblages. The locations of sampled trees were spaced with a distance of about 3–20 m between each other.

### Spider sampling

From 2005 to 2007, beech trees were fogged for spider collection using a Swingfog (SN-50) machine, which emits a natural pyrethrum insecticide that is not toxic to terrestrial vertebrates. Each sampled tree was fogged three times during a given fogging event. As spiders fell from the trees, they were collected on plastic sheets covering 90% of the fogged tree's crown vertical projection area on the ground (46 $m^2$ sheets for old-growth and mature beeches and 6 $m^2$ sheets for young beeches). Fogging began at daybreak and lasted approximately 10 minutes at every fogging time with an interval of 50 minutes. Spider samples were collected from the plastic sheets two hours after the completion of the three foggings. A total of 33 beech trees produced no spider samples after fogging. Subsequently, we had assemblages of 86, 101, and 102 trees in the old-growth, mature and young stands, respectively. The spider samples were stored in bags with 70% alcohol for transport to the laboratory, where specimens were identified to the species level using species-specific attributes of the palpal organ or epigynum according to *Heimer & Nentwig (1991)* and *Roberts (1996)*. Specimens were named according to the nomenclature suggested by *World Spider Catalog (2017)*.

To evaluate inventory completeness, the non-parametric estimator of species richness, Jack1, which represents species richness expected to be present in communities with highly

heterogeneous sample coverage, was computedin R version 3.2.3 (*R Development Core Team, 2017*) using the 'vegan' package (*Oksanen et al., 2015*).

## Environmental and spatial variables

Five architectural traits were measured for each individual tree, including diameter at breast height (DBH), total height ($H_t$), clear-pole height ($H_{cp}$), crown radius ($R$), foliage cover (FC) and canopy volume (CV), as indicators of habitat structure available to spiders. Clear-pole heightis the distance from underground to the lowest branch of an individual tree. Canopy volume is estimated by crown radius multiplied by canopy depth ($H_t$–$H_{cp}$). These five architectural traits varied greatly between trees and comprised a clear environmental gradient along which spiders were likely sorted into disparate assemblages.

Spatial variables were computed using Principal Coordinates of Neighbor Matrices (PCNM), a spatial analysis method for the detection of spatial trends across a range of scales (*Borcard et al., 2004*). First, the spatial coordinates of each sampling tree wereused to build a Euclidean distance matrix. Then, this matrix was truncated to the smallest distance that keeps all trees connected in a single network, which corresponds to the maximum distance between two successive sampling trees in one-dimensional studies. Finally, a Principal Coordinate Analysis (PCoA) was conducted on the Euclidean distance matrix to generate 23 PCNM eigenvectors with positive eigenvalues. These PCNM variables were then used as explanatory variables to analyze the spatial variation of the spider community composition.

## Classification of spiders with different dispersal capacity

Dispersal capacity varies widely among spider species and is tightly connected with ballooning propensity (*Jiménez-Valverde et al., 2010*). Previous studies have established ballooning propensity as the main method for evaluating dispersal capacity in spiders (*Thomas, Brain & Jepson, 2003*; *Jiménez-Valverde et al., 2010*). We assembled ballooning propensity information for all species encountered in this study from the existing literature (*Bell et al., 2005*; *Schirmel, Blindow & Buchholz, 2012*). Then we arranged the species into two categories according to their ballooning propensity: high- (where both adults and juveniles frequently balloon) and low- (where adults occasionally balloon but juveniles rarely balloon) vagility.

## Data analysis

We utilized the Bray-Curtis dissimilarity index (*Bray & Curtis, 1957*) to calculate spider compositional dissimilarities between trees using abundance data. We subsequently addressed whether the observed species turnover differed from the random expectation by comparing the observed values with 1,000 values generated by a null model. The null model randomized the community matrix while maintaining the observed species incidence across trees and the species richness in each tree (*Gotelli, 2000*). In doing so, we removed neutral sampling effects so as to test for the effects of underlying structured forces (e.g., habitat specialization and dispersal limitation). The standard effect size (SES) of non-random underlying forces was calculated as the difference between the observed and randomly expected dissimilarity. A SES value greaterthan zero indicates an identified

effect of habitat specialization and/or dispersal limitation, and the converse indicates the predominance of biotic homogenization (e.g., the loss of rare species, widespread species invasion and/or the overwhelming occupancy of common species).

To avoid collinearity among tree architecture variables, we applied variation inflation factor (VIF) values to eliminate redundant environmental variables (i.e., $H_t$ and $H_{cp}$). After checking the length of the dominant gradient in species composition ($L = 2.7$), a redundancy analysis (RDA) was employed to examine the relationship between spider composition and non-redundant tree architecture variables. Forward selection (*Blanchet, Legendre & Borcard, 2008*) was used to identify explanatory variables that significantly explain the variation in species composition ($P < 0.05$ after 1,000 random permutations). This procedure was also carried out for spatial (PCNMs) variables. Thirteen PCNM eigenvectors (i.e., PCNMs 1–9, 12, 25, 29 and 23) were preserved by a forward selection procedure in RDA.

Variation partitioning (*Borcard, Legendre & Drapeau, 1992*) was then performed to quantify the proportion of the variation in spider community composition explained purely by tree architectural dissimilarity, spatial distance, and community-level dispersal capacity dissimilarity. Community-level dispersal capacity dissimilarity was defined as the differences in vagility group composition between two communities in this study. Variation partitioning was carried out through a series of partial RDAs. Because the unadjusted $R^2$ values were biased (*Peres-Neto et al., 2006*), the $R^2$ values were adjusted to account for the explanatory variables. Negative adjusted $R^2$ can be ignored (considered as null) for the ecological interpretation of the results. We then performed 9,999 permutations testing for significance of unique fractions (i.e., tree architecture, spatial distance, dispersal capacity or their interactions) to determine how overall variation in species composition is partitioned among contrasting sets of explanatory variables. To evaluate the potential bias caused by the unmeasured variables in each stand, we also implemented variation partitioning in individual stands.

The beta diversity for each of the two dispersal capacity categories described above was calculated using the pair-wise dissimilarity measure based on the Bray-Curtis dissimilarity index. The dispersion of mean pair-wise dissimilarity values was estimated for each class by bootstrapping to detect significant differences between classes. One thousand bootstrap iterations were run, and the probability of obtaining lower mean pair-wise dissimilarity values for the high-vagility class in pair-wise comparisons by chance was calculated empirically. Likewise, variation partitioning analyses of beta diversity along environmental and spatial gradients were conducted for each spider dispersal capacity class.

The pair-wise dissimilarity measure was calculated using the 'betapart' package in R (*Baselga et al., 2013*), whereas bootstrap iterations and comparisons were conducted using the 'boot' package (*Canty & Ripley, 2015*). The null model was run using the 'picante' package (*Kembel et al., 2010*); and PCNM eigenvectors were created using the 'PCNM' package (*Legendre et al., 2009b*). The forward selection, variation partitioning, and tests of significance of the fractions were computed in R version 3.2.3 using the 'vegan' package (*Oksanen et al., 2015*).

**Table 1 Observed species richness, estimated species richness using non-parametric estimators (Jack1) and inventory completeness of beech (*F. sylvatica*) stands.**

|  | Old-grow stand | Mature stand | Young stand | All stands pooled |
|---|---|---|---|---|
| Observed richness | 55 | 66 | 51 | 88 |
| Estimated richness | 75 | 90 | 61 | 99 |
| Inventory completeness (%) | 73.3 | 73.3 | 83.6 | 88.9 |

## RESULTS

A total of 7,880 adult spider individuals belonging to 88 species, 61 genera and 16 families were collected throughout the survey from 2005 to 2007. The species inventory completeness values for each stand and all stands pooled were generally high (>73%, Table 1). Therefore, we considered the inventories to be comparable among stands.

Across all possible tree pairs, about 66.4% of the Bray-Curtis spider compositional dissimilarity values were higher than expected by chance (Fig. 2). Only a small proportion of tree pairs had lower-than-expected dissimilarity. Three tree architectural variables (i.e., DBH, CV and FC) were identified as significant predictors of spider composition, in which 14% of total variation was accounted for (Fig. 3). Tree size (i.e., DBH and CV) defined the major axis of distinguishing the composition of spider assemblages, and FC characterized the second axis.

Variance partitioning revealed that 28.4% of the variation in spider assemblage dissimilarity was explained by tree architecture, spatial distance and dispersal capacity. A relatively large part of this variation was attributed to spatially structured environmental components (8.7%), while 3.7%, 5.9%, and 6.8% were attributed to pure environmental components, pure spatial components, and pure dispersal capacity, correspondingly (Fig. 1B and Table S1).

In each stand, the three factors jointly explained a similar proportion (27%) of variation in spider composition (Fig. S1). However, two differences were detected in comparison with pooled analysis: (i) the importance of spatially structured environmental components dramatically diminished because of lower environmental variation within a stand than across stands; and (ii) the importance of dispersal capacity varied greatly probably due to the compositional differences across three stands.

The mean pair-wise dissimilarity of the high-vagility spider class was slightly lower than the low-vagility spider class but without statistical significance ($F = 26.59$, $df = 999$, $P = 1$). Pure environmental and spatial variables accounted for a similar proportion of the variation in compositional dissimilarity for both high- and low-vagility spider groups. Spatially-structured environmental factors explained twice as much variation in compositional dissimilarity for the high-vagility spider group as that for the low-vagility spider group (Fig. 4 and Table S2).

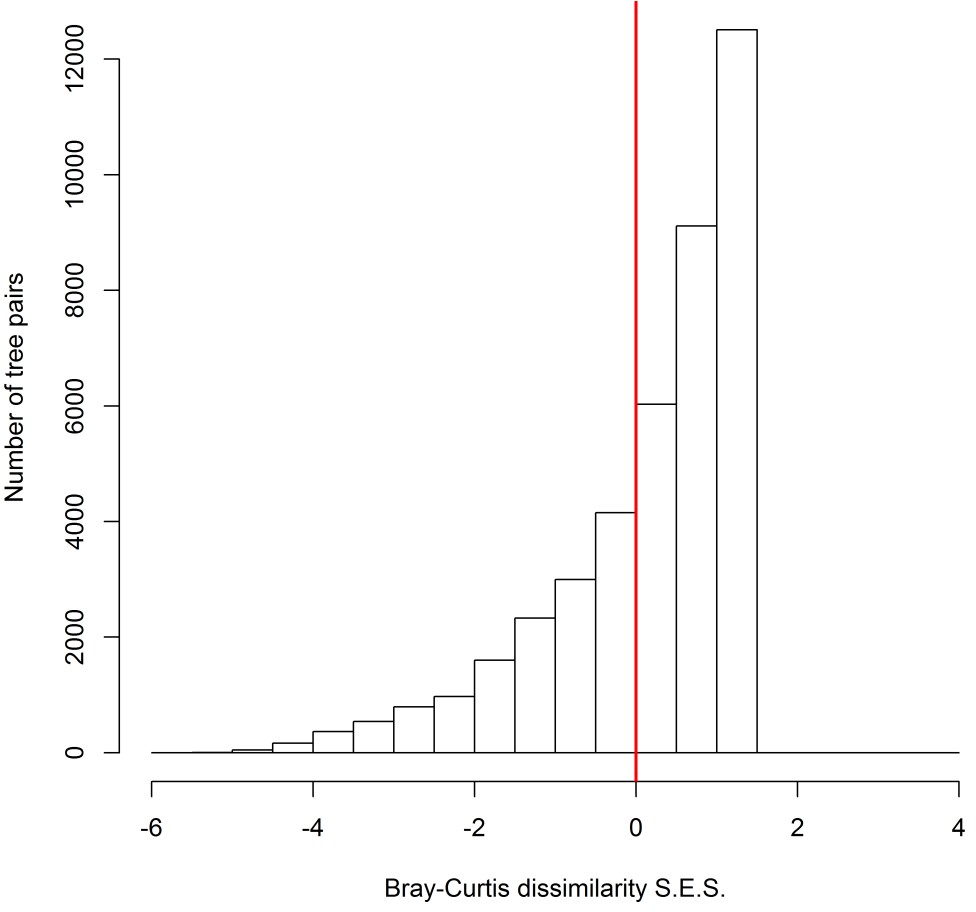

**Figure 2** **The standard effect size of non-random processes on spider compositional dissimilarity (i.e., Bray-Curtis index) between trees.** SES values above zero indicate that observed compositional dissimilarity is higher than expected by random assembly. The proportion of tree pairs with SES values above zero are 66.4%.

## DISCUSSION

Although it is difficult to sample the complete spider assemblage due to a large number of rare species collected, according to the Jack1 estimators, nearly 90% of the spider species expected to be present in the beech forest canopies were captured in this study and more than 73% of these species were observed in each stand (Table 1). These results suggest inventories were adequate in this study, thereby enabling us to accurately quantify spider assemblage diversity within the beech forest. Nevertheless, it is important to keep awareness of some shortcomings associated with fogging. For example, some dead spiders might be suspended by their silk 'safety lines' and thus not be collected (*Yanoviak, Nadkarni & Gering, 2003*); individuals with small body size might be trapped in minute droplets of spray in trees, or be blown away before being collected (*Noyes & Sadka, 2003*). In addition, due to daily air exchange patterns, fogging was mostly implemented at dawn and unable to
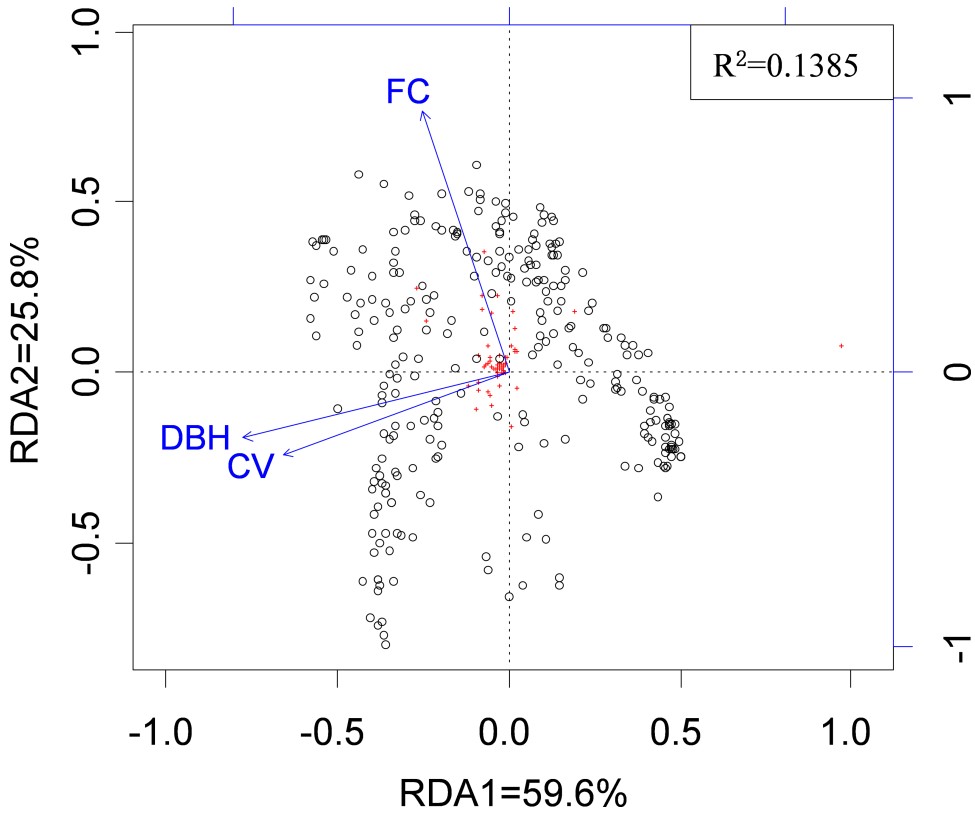

**Figure 3** **The relationship between spider species composition and environmental factors as revealed by RDA.** Circles represent 289 tree-defined spider assemblages, and red crosses represent 88 species. Arrows denote environmental factors: DBH, Diameter at the breast height; CV, canopy volume; FC, foliage cover.

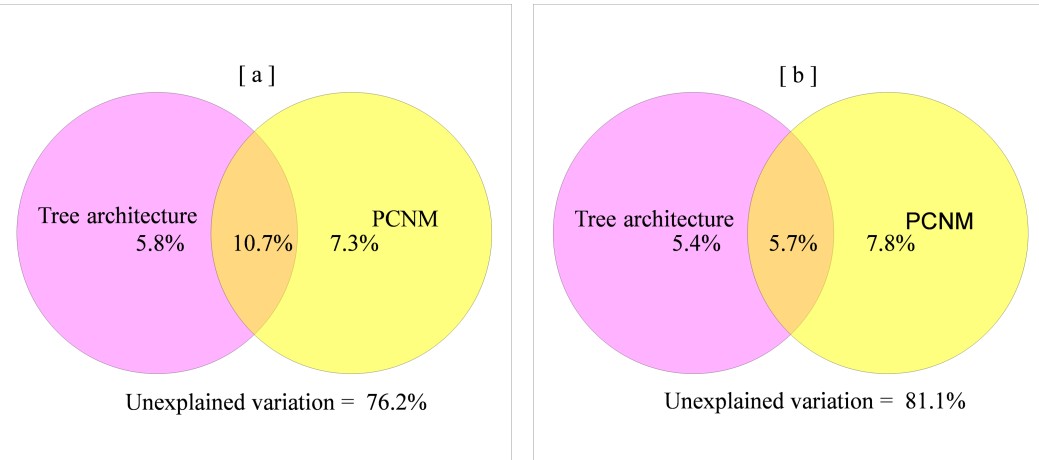

**Figure 4** **Relative influence of environmental and spatial distance on beta diversity patterns of the spider group with high (A) versus low (B) vagility.**

capture nocturnal species (*Hsieh, 2011*). Therefore, data collection via fogging could partly limit the generality of this study.

## Tree architecture as an environmental gradient

We found that trees with dissimilar DBH and CV, and to a lesser extent FC, supported more dissimilar spider assemblages. These factors explained about 14% of the variation in spider composition. Similarly, *Halaj, Ross & Moldenke (1998)* and *Halaj, Ross & Moldenke (2000)* showed that leaf density and branching structure regulated the abundance and distribution of arboreal spiders. Altogether, tree architecture comprises a relevantly environmental gradient along which arboreal spiders are sorted into disparate assemblages in terms of their niche specialization.

Several studies have identified vegetation structure as a key habitat signals of spider communities (*Carvalho et al., 2011*; *Gómez, Lohmiller & Joern, 2016*; *Rodriguez-Artigas, Ballester & Corronca, 2016*). Here we showed that lower-level variables of vegetation structure (i.e., individual-tree architectural traits) also influence arboreal spider composition. These results collectively endorse that non-trophic habitat heterogeneity within and among forest stands acts as a ''bottom-up template'' for structuring spider assemblages (*Halaj, Ross & Moldenke, 2000*).

## The importance of environmental constraints and dispersal limitation

The null model analysis showed that observed dissimilarity for most spider assemblage pairs was higher than expected by chance, indicating a higher overall species turnover rate than randomly expected across individual trees. This pattern in beta diversity probably reflects prominent effects of habitat specialization and/or dispersal limitation (*Tuomisto, Ruokolainen & Yli-Halla, 2003*; *Anderson et al., 2011*). In contrast, biotic homogenization appears to play a subordinate role, if any, in structuring spider assemblages in this study.

Our results showed that a combination of tree architecture, spatial distance and dispersal capacity explained a considerable proportion of the variation in beta diversity, and these three components represented a comparable proportion to each other. These results support the widely-held opinion that niche-based and dispersal processes are not mutually exclusive, but may work together to influence species diversity and coexistence (*Gilbert & Lechowicz, 2004*; *Legendre et al., 2009a*).

## Influences of dispersal capacity on local beta diversity

Our results indicate that dispersal capacity also plays a role in shaping beta diversity as spatial distance, suggesting that both of these factors should be considered when evaluating the importance of dispersal limitation. Interestingly, we found the mean pairwise compositional dissimilarity between sampling units (i.e., individual trees) was similar between two dispersal capacity classes within a confined spatial range. Furthermore, we found little support for heavy spatial effects on beta diversity patterns of low-vagility spiders, suggesting that spatial autocorrelation in spider composition is not necessarily higher for low-vagility spiders. These findings suggest that the negative relationship between species turnover level and dispersal capacity found at large geographic scales

(*Jiménez-Valverde et al., 2010*; *Rodriguez-Artigas, Ballester & Corronca, 2016*) may not be readily generalized at local scales.

Notably, we discovered that spatially structured environmental factors represented a far more important determinant of beta diversity for the high-vagility spider group than that for the low-vagility group. Due to the more pronounced spatial structure of environmental variation at the broad scale (pooled analysis in Fig. 1) than those at the fine scale (individual stands in Fig. S1), it seems that the distribution of high-vagility spiders is more responsive to the broad-scale environmental variation. In other words, high-vagility helps promote habitat tracking (*Araújo & Pearson, 2005*). In contrast, low-vagility spiders appear to live in a narrow niche because they are unable to adapt to broad-scale environmental variation ($r = 0.126 \pm 0.124$, polychoric correlation test between species frequency and vagility classes). As a result, dispersal capacity interacts with niche breadth and spatial scale to regulate spider beta diversity.

## CONCLUSIONS

Our results suggest both niche specialization, along with the environmental gradient defined by tree architecture, and dispersal limitation are responsible for structuring arboreal spider assemblages. In light of the importance of tree architecture, especially tree size that is a relevant environmental gradient for sorting arboreal spider species, stand structural heterogeneity is of great value in conserving spider biodiversity. For example, multi-cohort forests can provide more varied niches to promote species coexistence. In addition, less vagile spiders may be seen as rare species and be more affected by stochasticity. In this way, their higher conservation value is justified, but followed by a challenge of conserving these species with low predictability.

It should be noted that the reason for a large amount of variation ($\sim$72%) in compositional dissimilarity of spider community remain unclear, which may be explained by the influence of stochastic processes, in which population dynamics are primarily driven by ecological drift and habitat independence (*Legendre et al., 2009a*). Additionally, some important variables, such as food sources and micro-climate factors that may be responsible for the assembly of spider communities in these trees, were not measured in this study, which may also contribute to the large variation.

## ACKNOWLEDGEMENTS

We thank Yu-Lung Hsiehand Karl Eduard Linsenmair, who provided the dataset for this study. We are grateful to Fangliang He and Chengjin Chu of the SYSU—Alberta Joint Lab for Biodiversity Conservation, who provided helpful suggestions in conducting and publishing this study. Xinghua Suiis appreciated for his assistance in data analysis. We thank Christine Verhilleat the University of British Columbia for technical editing on an early version of the manuscript.

### Funding

The authors received no funding for this work.

### Competing Interests

The authors declare there are no competing interests.

### Author Contributions

- Qiongdao Zhang conceived and designed the experiments, performed the experiments, analyzed the data, contributed reagents/materials/analysis tools, prepared figures and/or tables, authored or reviewed drafts of the paper, approved the final draft, data collection.
- Wei Shi conceived and designed the experiments, analyzed the data, contributed reagents/materials/analysis tools, authored or reviewed drafts of the paper, approved the final draft.
- Hua Wu performed the experiments, analyzed the data, prepared figures and/or tables, authored or reviewed drafts of the paper, approved the final draft.
- Dong He conceived and designed the experiments, authored or reviewed drafts of the paper, approved the final draft.
- Cong Chen performed the experiments, contributed reagents/materials/analysis tools, authored or reviewed drafts of the paper, approved the final draft.

### Data Availability

The raw data and R script code are provided in the Supplemental Files.

### Supplemental Information

Supplemental information for this article can be found online at http://dx.doi.org/10.7717/peerj.5596#supplemental-information.

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
