# Peer review of "Local-scale determinants of arboreal spider beta diversity in a temperate forest: roles of tree architecture, spatial distance, and dispersal capacity"

_PeerJ, doi:10.7717/peerj.5596_

## Round 0.1 · original submission · Major Revisions

Both reviewers thought the submission well written and methodologically sound, but they also raised some concerns about survey background, data analysis and your presentation (pay attention to these). Both of them provided very helpful suggestions for you revising the manuscript. I suggest you should adopt suggestions which can improve your manuscript and revise the submission carefully.

Reviewer 1 ·

Basic reporting

This manuscript by Zhang et al. presents a large dataset on arboreal spider species in Germany. Overall, it is well organized and well written. However, there are some potential problems in the data analysis approaches and the explanations on the results (see my comments below).

Experimental design

(1) The data were collected in different forest types in Germany, including old-growth beech, mature beech, and young beech. But I cannot find the comparisons among these three types, which should be more interesting for the readers. To mix all the data together for the current analysis, it leads to the results are really hard to be explained.
(2) Physical environment: Does the survey collect any physical environmental variables? I didn’t find any analysis on that. The authors mixed biotic and abiotic environment together in this MS. I think that it is important to detangle them.

Validity of the findings

(1) Spatial variables: The authors use the spatial variables to link with dispersal limitation and neutral theory. I find that it could be really weak when you didn’t consider any physical environment.
(2) Fine scale vs. broad scale: Don’t understand how the authors classify fine and broad scale. How large could be defined at the broad scale? I would like to see the clear definition on them.

Additional comments

(1) Abstract Line 21: “We surveyed”? Did the authors really do the field survey? As I read from the text, the data were collected by other people. I suggest that the authors rewrite the similar descriptions to make it clear.
(2) Lines 246-247 “some shortcomings associated with fogging”: please list the shortcomings.
(3) Lines 268-269: “biotic homogenization processes”: For my understanding, biotic homogenization is a pattern. How could it be the “processes”? What is the underlying mechanism(s) to drive the homogenization? This question is something you need to discuss here.
(4) In the figures, you use both “stand structure” and “environmental variables”. What is the difference between them?

Reviewer 2 ·

Basic reporting

Major Concerns:

1. Although all of the information is in the text, the structure of the results and methods do not correspond and neither are directly connected to the three main questions presented in the last paragraph of the introduction. This makes it difficult to compare these sections and determine how the observation in this project relate to the central questions of the manuscript. The manuscript would greatly benefit from explanations of how each analysis conducted specifically addresses one of the major question from the introduction.

2. The introduction as currently structured is hard to follow and does not intuitively lead the reader to the three questions underpinning the manuscript. The authors should reorganize the introduction to focus on how this project fits within a larger ecological and biological framework and specifically focus on how the three questions are important to understanding beta diversity patterns. Spiders are a useful model for beta diversity studies but should be secondary to this wider framework. Rodriguez-Artigas et al. 2016 provides a good model for reframing this paper.

3. The sixth paragraph of the introduction (Lines 78 – 86) does well explaining the contrast in environmental versus spatial factors but does not fully explain what “spatially-structured environmental variables” represent. These different terms are central to the overall story of the paper. Presenting direct examples of each would provide more clarity.

4. The discussion could be reduced by about 25%. This could mostly be accomplished by removing the reiterated results in the discussion section and by focusing on the implications of these results on the larger understanding of patterns of spider beta diversity. Several of the smaller paragraphs could be collapsed into more concise single paragraphs.

Minor Issues:

1. The studies cited establishing that structural variability in vegetation influences spider communities are from coastal and grassland environments (Line 62). Further support is needed to justify how the variables of tree architecture that were measured in the current study (Lines 136-143) influence spider communities. Specifically, authors should provide citations for how and why “architecture of individual trees serves as the immediate habitat template for spider assemblages (Lines 65-66).” There is ample evidence of the role crown architecture plays in shaping spider assemblages (Halaj et al 2000) and other arthropod communities that is overlooked in the currently in the introduction.

2. Citations are needed for the information presented in the first paragraph in the methods (Lines 107 – 112).

3. Citations are needed for methodology and claims presented concerning null models (Lines 162-171).

4. Figure 3 and Figure 5 refer to “environmental (tree architecture)” as a variable in the caption and throughout the text. However, this is called “Stand structure” in the figures themselves. Consistent terminology should be used.

5. Figure 4 shows that spatial components were slightly greater at fine scales compared to broad scales; however, the text in the manuscript (Line 229-230) indicates the opposite pattern. It is unclear from the discussion which is correct.

6. Figure captions are cut off in the document provided to the reviewers. This presented some issues with understanding the figures but is likely just an uploading error. This should be addressed before final submission.

Experimental design

Major Issue:

1. It seems highly unlikely that trees in each of these three forest stands would all be regularly distributed at approximately 5m distances (Lines 116-117). It is not clear from the current methods how the trees were selected to achieve this distribution. Additionally, it is not clear from the description if the three tree stands were contiguous or if there is spatial separation among these stands. This is important because it could impact the PCNM analyses and all of the final results presented. Additionally, the introduction of the three forest stands used to provide a wider variety of tree architecture also introduces a confounding variable of stand age that should be considered at the very least in the discussion if not via statistical means.

Validity of the findings

no comment

Additional comments

The manuscript describes the variability in beta diversity of spiders collected from beech trees in the Würzburg University Forest specifically focusing on the roles of tree architecture, tree spatial distributions, and spider dispersal capacity. Spider compositional dissimilarity between trees was higher than expected by random chance and spatially-structured environmental factors contributed most to variation at broad scales compared to fine scales. However, >78% of variation in beta diversity was not explained by the measured variables. In contrast with previous work, spatial factors did not dramatically effect beta diversity in low vagility spider species. The manuscript is methodologically sound, technically well written and provides results that should be of interest to a wide ecological audience. However, as currently structured it is difficult to connect the three main questions of the paper to the methods and results presented. Additionally, there are aspects of the field methodology that need further clarification. The specific concerns are outlined in the specific sections.

---

## Round 0.2 · Minor Revisions

I agree with the reviewers that the authors did not successfully deal with some major concerns in the first round of review. I suggest the authors should really and carefully think about these concerns especially that related to three different forest types as well as clarifying and reformatting the manuscript. We would like to receive your improved revision soon. And I am sorry for my delayed response due to a leave for some time.

Reviewer 1 ·

Basic reporting

The revised version of the manuscript has been greatly improved.

Experimental design

I don’t think that the authors address my suggestions well. I put my comments in the original manuscript here and add my new comments after those.
(1) The data were collected in different forest types in Germany, including old-growth beech, mature beech, and young beech. But I cannot find the comparisons among these three types, which should be more interesting for the readers. To mix all the data together for the current analysis, it leads to the results are really hard to be explained.
New comment: The authors responded that “this pooled analysis served for the same end as the comparisons among forest types”. I am fine with the pooled analysis, but I think that it is important to go back to the raw idea on the comparison of these three different forest types. You need to describe whether or not and why there are some different species composition, and what the results mean for species conservation or forest management.

(2) Physical environment: Does the survey collect any physical environmental variables? I didn’t find any analysis on that. The authors mixed biotic and abiotic environment together in this MS. I think that it is important to detangle them.
New comment: I don’t like the term “biotic environmental variables”. It is a strange usage. Do you see any one used this term in the literature? There might some existed, but I am sure that it is really rare. Why not use “tree architecture” directly? I will suggest you to use this term through the whole manuscript.
In addition, you mention that “to disentangle the roles of tree architecture, spatial distance, and dispersal capacity on spider turnover” in the abstract. I am wondering which variable is directly related to dispersal capacity.

Validity of the findings

(2) Fine scale vs. broad scale: Don’t understand how the authors classify fine and broad scale. How large could be defined at the broad scale? I would like to see the clear definition on them.
New comment: The question is still not clear for me. Usually, ecologists use “broad scale” for regional or national scale analysis. The current study is on a local scale. Since it is not a main focus of this paper. I will suggest to remove all the comparisons between fine and “broad” scales.

Additional comments

(1) In the figures, you use both “stand structure” and “environmental variables”. What is the difference between them?
New comment: Does “stand structure” equal to “tree architecture”? If so, why not use the same term through the whole manuscript?

Reviewer 2 ·

Basic reporting

This is the second time that I have reviewed this paper and upon reading it again, I still admire the robust statistical approaches the authors took to addressing spider beta diversity within this forest. However, many of the same issues I had in my first review remain and new issues are now apparent with the addition of the map provided in the response to reviewers. The specific details are outlined below:

Major Concerns
The overall structure of the paper fails to present the information in a clear and concise manner. Within the response to reviewers, the authors provided very clear predictions with a very clear outline of how every test in the methods relates to a specific prediction. This should serve as a template for be how the manuscript should be reformatted.

Additionally, it is unclear why spider vagility cannot be described in terms of dispersal, allowing for the paper to be framed within a larger ecological content rather than a spider-centric focus.

Finally, the discussion still unnecessarily reiterates results rather than focusing on the implications of the results. Ultimately, the length of the discussion can still be reduced by about 25% as it is the exact same size as it was in the first submission.

Experimental design

Both reviewers point out that the addition of three forest stands, for the purpose of expanding the variation in tree architecture, also introduces a confounding variable of stand age. It is insufficient to assume that differences in tree architecture is the best explanation for differences in spider beta diversity considering the wide array of differences in forests of different ages. Additionally, the map provided in the response to reviewers also makes it clear that these stands are spatially distinct confounding measures of tree architecture with not only stand age but also physical location. Further consideration needs to be given to the role of stand age and the physical separation among these distinct forest stands before this manuscript is ready for publication.

Validity of the findings

No comment

---

## Round 0.3 · accepted · Accept

After two rounds of peer review and revision, I think the manuscript has been greatly improved. I am happy to accept it for publication. Congratulations!

# Reviewer 1 ·

Basic reporting

The current version of the manuscript has been greatly improved. I am happy with the revision. I think that it is ready to publish.

Experimental design

No comments.

Validity of the findings

No comments.

Additional comments

No comments.